# Mass Spectrometry-Based Proteomic Analysis of Potential Host Proteins Interacting with GP5 in PRRSV-Infected PAMs

**DOI:** 10.3390/ijms25052778

**Published:** 2024-02-28

**Authors:** Wen Li, Yueshuai Wang, Mengting Zhang, Shijie Zhao, Mengxiang Wang, Ruijie Zhao, Jing Chen, Yina Zhang, Pingan Xia

**Affiliations:** 1College of Veterinary Medicine, Henan Agricultural University, Longzi Lake 15#, Zhengzhou 450046, Chinazsj1002935527@163.com (S.Z.); wmxjya@163.com (M.W.);; 2College of Life Science, Henan Agricultural University, Longzi Lake 15#, Zhengzhou 450046, China

**Keywords:** PRRSV, GP5, LC-MS/MS, protein–protein interaction, antigen processing and presentation

## Abstract

Porcine reproductive and respiratory syndrome virus (PRRSV) is a typical immunosuppressive virus causing a large economic impact on the swine industry. The structural protein GP5 of PRRSV plays a pivotal role in its pathogenicity and immune evasion. Virus–host interactions play a crucial part in viral replication and immune escape. Therefore, understanding the interactions between GP5 and host proteins are significant for porcine reproductive and respiratory syndrome (PRRS) control. However, the interaction network between GP5 and host proteins in primary porcine alveolar macrophages (PAMs) has not been reported. In this study, 709 GP5-interacting host proteins were identified in primary PAMs by immunoprecipitation coupled with liquid chromatography-tandem mass spectrometry (LC-MS/MS). Bioinformatics analysis revealed that these proteins were involved in multiple cellular processes, such as translation, protein transport, and protein stabilization. Subsequently, immunoprecipitation and immunofluorescence assay confirmed that GP5 could interact with antigen processing and presentation pathways related proteins. Finally, we found that GP5 may be a key protein that inhibits the antigen processing and presentation pathway during PRRSV infection. The novel host proteins identified in this study will be the candidates for studying the biological functions of GP5, which will provide new insights into PRRS prevention and vaccine development.

## 1. Introduction

Porcine reproductive and respiratory syndrome (PRRS) is a viral disease caused by PRRS virus (PRRSV) with serious reproductive disorders in sows and respiratory symptoms in piglets, leading to significant economic losses in the pig industry worldwide. Genetic diversity, immunosuppression, and the antibody-dependent enhancement (ADE) effect of PRRSV pose additional challenges for PRRS prevention and control [1,2,3,4,5,6,7,8]. Although vaccination is the most effective method to prevent PRRSV infection, the efficacy of PRRSV vaccines remains unsatisfactory [9,10,11]. It is reported that neutralizing antibodies (NAs) play a significant role in protecting pigs against PRRSV infection, but the delayed response of NAs leads to the low effectiveness of PRRSV vaccines [12,13]. Thus, it is essential to explore the biological processes and the molecular mechanisms underlying PRRSV infection. 

PRRSV is an enveloped single-stranded positive-sense RNA virus in the *Arteriviridae* family, its viral proteins including structural proteins GP2a, GP2b, GP3-GP5, GP5a, M, and N, as well as non-structural proteins (NSPs) nsp1α, nsp1β, nsp2-6, nsp7α, nsp7β, and nsp8-12 [14]. Of these, structural protein GP5 serves as a major virulence determinant. It is the preferred protein for developing new vaccines due to its important immune domains such as neutralizing epitope, antigenic determinants, and glycosylation sites [15,16,17,18,19,20]. Moreover, GP5 plays a vital role not only in PRRSV infection, cell binding, and immune evasion, but also in cell apoptosis [21,22,23,24,25]. However, as the main target antigen of PRRS vaccine, the induction of NAs induced by GP5 occurs late, resulting in reduced immunological protection [26]. Therefore, understanding the interaction between GP5 and host proteins could provide new ideas for exploring PRRS vaccines targets and provide the scientific basis for the prevention and control of PRRS. 

Protein interactions between viral proteins and host proteins are of great importance to the viral entry, replication, and suppression of host-immune responses [27,28,29]. Therefore, the study of virus–host interactions promises significant advancements in controlling viral infection. As is well known, under in vitro condition, PRRSV can propagate in the African green monkey kidney cell line MA-104 and its derivatives, including MARC-145 cells [30]. At present, several studies have identified GP5-interacting cellular partners in MARC-145 cells, such as CD169 (Sn), MYH9, ATP5A, GAPDH, Snapin, and so on [30,31,32,33,34,35,36]. Thereinto, CD169 and MYH9 involve in PRRSV internalization, and GAPDH facilitates PRRSV replication. However, pigs are the natural hosts for PRRSV. Furthermore, PRRSV has a tropism for monocyte lineage cells, especially differentiated macrophages, and primarily replicates in porcine alveolar macrophages (PAMs). Thus, it is better to simulate the natural infectious status in PRRSV-infected primary PAMs than in MARC-145 cells. However, the interaction network between GP5 and host proteins in primary PAMs has not been reported. 

In this study, we identified potential GP5-interacting host proteins in primary PAMs using liquid chromatography-tandem mass spectrometry (LC-MS/MS)-based proteomics approach. With analyzing the functions of these GP5-interacting proteins and validating the effect of GP5 on these proteins, we try to summarize important information about GP5 biological functions and PRRSV pathogenesis and provide a valuable basis for PRRS control and vaccine development.

## 2. Results

### 2.1. Identification of Potential GP5-Interacting Proteins in PAMs

To understand the functions of GP5, we identified GP5-interacting partners in PRRSV-infected PAMs by co-immunoprecipitation (Co-IP) assay combined with LC-MS/MS. As shown in Figure 1, western blotting analysis validated that the GP5 band was present only in the GP5-IP sample, which indicated that GP5 was successfully pulled down by IP. Proteins detected in IgG-IP samples were considered nonspecific; therefore, the differentially expressed bands between the lane of GP5-IP and IgG-IP were excised from the gels and then identified by LC-MS/MS. In total, 1586 proteins were identified that interact with GP5. To reduce the probability of false peptide identification, each protein involved at least two unique peptides, and peptides with average Mascot scores ≥ 14 were counted as identified. By subtracting 877 unidentified binding proteins, 709 proteins were identified as potential GP5-interacting proteins in PAMs. Details of the identified proteins are shown in Appendix A.

### 2.2. Gene Ontology (GO) Annotation of GP5-Interacting Proteins

To further investigate the functional processes associated with GP5-interacting proteins, GO cluster analysis was performed to provide relevant information about biological process (BP), cellular component (CC), and molecular function (MF). As shown in Figure 2A, most of these proteins were linked to translation, protein transport, and protein stabilization in BP. In terms of the CC annotation, these proteins were primarily localized in the cytoplasm, nucleus, and mitochondrion (Figure 2B). For MF, the top-ranked categories were the binding, activity, and structural constituent of ribosome (Figure 2C).

### 2.3. Kyoto Encyclopedia of Genes and Genomes (KEGG) Pathway Annotation of GP5-Interacting Proteins

To comprehensively understand the biological pathways associated with GP5-interacting proteins, we conducted KEGG analysis on these proteins. Based on the results of KEGG pathways analysis, a total of 48 pathways was identified, and the complete KEGG pathway annotation information can be found in Appendix A. It is worth noting that the Top 20 KEGG pathways were mostly divided into metabolism, genetic information processing, and cellular processes. In addition, GP5-interacting proteins were mainly enriched in the pathways belonged to the metabolic pathways (ssc01100), ribosome (ssc03010), spliceosome (ssc03040), protein processing in the endoplasmic reticulum (ssc04141), and nucleocytoplasmic transport (ssc03013) (Figure 3).

### 2.4. Protein–Protein Interaction (PPI) Network Analysis of GP5-Interacting Proteins

Previous studies have demonstrated that PRRSV nsp4 modulates the host response by disrupting the SLA-I antigen presentation pathway [37]. Interestingly, as shown in Appendix A, KEGG pathways enrichment data suggest a potential association between GP5 and antigen processing and presentation, which were previously unknown pathways. Therefore, to comprehensively understand the mechanisms of antigen processing and presentation induced by PRRSV, we selected 163 proteins from the Top 20-ranked categories in BP for subsequent protein-protein interaction (PPI) network analysis. These proteins were involved in various aspects of antigen processing and presentation, such as proteins translation, transport, and stabilization. The PPI networks among them were established by the STRING database (confidence score > 0.7 (high confidence) and PPI enrichment *p* < 1.0 × 10^−16^). Subsequently, the PPI network was visualized using Cytoscape Software (version 3.7.0). The analysis revealed that the network had 163 nodes and 900 edges, and the disconnected nodes were excluded from the network. Significantly, GP5-interacting proteins were centrally distributed in three PPI network clusters, which were associated with protein translation, protein transport, and protein processing, respectively (Figure 4).

### 2.5. Validation of the Interaction between GP5 and Identified Host Proteins

Most of the proteins identified in this study were previously unknown as interacting partners of GP5. Considering that PRRSV is an immunosuppressive pathogen and GP5 contains neutralizing epitopes, we selected eight antigen processing and presentation related proteins, including TAP1, TAP2, CANX, CALR, SLA-II, PSMB7, VPS16, and VPS18. Then, we verified the interactions between them and GP5 by Co-IP assay and indirect immunofluorescence assay (IFA). The Co-IP assays displayed that GP5 was only detected in the over-expressing cells (Figure 5A). The confocal microscope analysis revealed that there were colocalizations between GP5 and the selected proteins (Figure 5B). The results showed that all the selected proteins could interact with GP5, providing a certain confidence for the identification of GP5-interacing proteins from the LC-MS/MS data.

### 2.6. PRRSV May Inhibit the Progress of Antigen Processing and Presentation via GP5

To determine the effect of PRRSV on host proteins, we quantitatively assessed protein levels in PRRSV-infected PAMs by a tandem mass tag (TMT)-based quantitative proteomic approach. Results displayed that the protein levels of identified GP5-interacting proteins TAP1, TAP2, CANX, CALR, SLA-DRA, and SLA-DMA were down-regulated (Figure 6A and Appendix A). Subsequently, the protein levels of SLA-II in PRRSV-infected PAMs were detected by western blotting assay. As shown in Figure 6B, compared to the Mock group, the protein levels of SLA-II were significantly down-regulated in the PRRSV group. The results indicate that PRRSV may inhibit the progress of antigen processing and presentation. 

To further evaluate the effect of GP5 on antigen processing and presentation, recombination vector Flag-SLA-II was co-transfected with Myc-GP5 into HEK293T cells. The western blotting assay showed that over-expression of GP5 significantly decreased the protein levels of SLA-II (Figure 6C). The results reveal that GP5 may inhibit the progress of antigen processing and presentation through down-regulating the expression of SLA-II. The above results suggest that GP5 may be a key protein that inhibits the antigen processing and presentation pathway during PRRSV infection.

## 3. Discussion

PRRSV could evade host innate immunity, leading to a delayed response of NAs and immunosuppression [8,12,22]. PRRSV structural protein GP5 has a neutralizing epitope and plays an important role in the pathogenicity and immune evasion of PRRSV [16,20,22]. To further understand the biological function of GP5, we identified 709 potentially GP5-interacting host proteins in PAMs by Co-IP assay combined with LC-MS/MS-based proteomics approach (Figure 1). Previous studies have identified GP5-interacting cellular partners in MARC-145 cells [33,34]. However, the main target for PRRSV is primary PAMs during infection [38]. Therefore, it better simulate the natural infectious status in PRRSV-infected primary PAMs than in MARC-145 cells. More importantly, novel GP5-interacting cellular partners may be found in primary PAMs.

To learn more about the unknown molecular functions of GP5, we performed a bioinformatic analysis on these identified interacting host proteins of GP5. It is noteworthy that several identified potentially host proteins belong to regulatory peptides and distinct protein forms. Peptides are derived from protein hydrolysis and possess unique structures and biological functions. Additionally, different amino acid compositions result in distinct properties among various protein forms. For instance, subunits PMSB7 and PMSB6 within the 26S proteasome exhibit trypsin-like and caspase-like enzymatic activities, respectively. Importantly, these proteins were all identified by unique peptides. Therefore, we thought they could be employed in inferred pathways [39]. It has been reported that GP5-interacting cellular proteins in MARC-145 cells are mainly involved in cellular process, biological regulation and metabolic process, and the major subclasses enriched in molecular function included binding, catalytic activity, and structural molecule activity [33]. Moreover, Oh and Lee [40] found that GP5a affects proteins involved in diverse cellular functions, including cell growth, cytoskeleton networks and cell communication, metabolism, protein biosynthesis, RNA processing, and transport. Consistent with these findings, our study also revealed significant enrichment of GP5-interacting cellular proteins in translation, protein transport, and protein stabilization in PAMs (Figure 2). These results demonstrate that PRRSV may utilize GP5 to manipulate multiple cellular processes such as protein synthesis, protein transport, and protein degradation. 

Previous studies have reported that GP5 could induce cell apoptosis, inhibit the phosphorylation of interferon regulatory factor-3 (IRF-3) in the IFN signaling pathway, and exploit the glycolytic activity of GAPDH to control PRRSV replication [34,41,42]. Likewise, our KEGG pathway analysis revealed that multiple GP5-interacting proteins are also enriched in apoptosis and glycolysis pathways. Furthermore, CD163, CD169 (Sn), MYH9, ATP5A, GAPDH, and Snapin have been reported to interact with GP5 and play a role in viral replication [30,31,32,33,34,35,36]. Similarly, MYH9, ATP5A, and GAPDH were identified in our results (Appendix A). Puzzlingly, CD163 and CD169, as the receptors of PRRSV, were not detected in this study. According to the reports, CD163 and CD169 are transmembrane proteins [30]. Different from cytoplasmic proteins, it is challenging to efficiently extract and isolate transmembrane proteins due to their membrane-spanning domains and hydrophobicity [43,44]. We thus suspect that some of these transmembrane proteins, such as CD163 and CD169, may not have been efficiently extracted and isolated; as a result, these low-abundance proteins could not be detected. Moreover, the samples analyzed by LC-MS/MS were obtained from differentially expressed bands in gels rather than whole cell lysates, which could explain why certain proteins remained undetected. Additionally, the inherent limitations of protein identification by mass spectrometry, such as missed and non-specific cleavage of proteins during trypsin treatment, may also be responsible for the absence of these proteins [45].

Notably, according to the complete KEGG pathway annotation information (Appendix A), we discovered a potential link between GP5 and antigen processing and presentation, which has not been previously reported. The KEGG result in Figure 3 displays only the Top 20 pathways, while the antigen processing and presentation pathway does not rank among them. This could be attributed to the fact that the samples analyzed by LC-MS/MS were obtained from the differentially expressed bands in gels rather than whole cell lysates, which resulted in some antigen processing and presentation pathway related proteins being undetected. However, it is important to note that antigen processing and presentation play a crucial role in inducing specific immune responses such as antibody production or T-cell activation for virus clearance [46,47]. Previous studies reported that GP5 could be used as a target for NAs [16,17]. Therefore, we speculated that GP5 may inhibit the immune response by interacting with processing and presentation related proteins during PRRSV infection. Thus, research on interactions between GP5 and antigen presentation related protein is helpful to clarify the molecular mechanism underlying PRRSV-induced immunosuppression. 

In this study, we selected eight identified host proteins, TAP1, TAP2, CANX, CALR, SLA-II, PSMB7, VPS16, and VPS18, involved in antigen processing and presentation pathways, to verify the interactions between them and GP5 by Co-IP assays and IFA. Transporter associated with antigen processing (TAP) protein is a heterodimeric protein comprised of TAP1 and TAP2, which utilizes ATP to transport cytosolic peptides into the ER across its membrane. In the ER, together with CANX and calreticulin, CALR forms the peptide loading complex (PLC), which directs peptides onto major histocompatibility complex class I (MHC-I) molecules, subsequently transported to the cell surface for CD8^+^ T cells recognition [46,48]. Major histocompatibility complex class II (MHC-II) molecules typically bind peptides generated by lysosomal proteolysis in the endocytic and phagocytic pathways, which are subsequently transported to the cell surface and recognized by CD4^+^ T cells [49]. PSMB7 is a subunit of proteasome, which plays a key role in the generation of peptides presented by MHC-I molecules [50]. Autophagy is critically involved in antigens presentation, and VPS16 and VPS18 are required for fusion of autophagosome with lysosomes [51,52]. Results showed that all these selected proteins could interact with GP5 (Figure 5). The higher validation percentage provides a certain confidence for these potential GP5-interacting proteins in LC-MS/MS data. Our study considerably expands the knowledge on novel functions of GP5. However, it should be noted that these GP5-interacting proteins might directly interact with GP5 or indirectly via protein complexes. Therefore, extensive validation is necessary for further research.

According to the reports, PRRSV has evolved various strategies to disrupt the host antiviral system and provide favorable conditions for survival [53]. Therefore, understanding the interaction between GP5 and host proteins could lay the foundation for exploring the escape mechanism of PRRSV. Recent research has suggested that many viruses have developed appropriate countermeasures to evade immune surveillance by interfering with antigen processing and presentation pathway. For example, Herpesviruses could prevent transporter associated with TAP-dependent peptide translocation into the endoplasmic reticulum lumen by targeting the TAP proteins, thus evading immune surveillance [48]. Human immunodeficiency virus type 1 (HIV-1) Nef interacts with low molecular mass protein 7 (LMP7) to attenuate immunoproteasome formation and the MHC-I antigen presentation [54]. SARS-CoV-2 ORF7a physically associates with MHC-I, thereby preventing the assembly of the MHC-I peptide loading complex and inhibiting the presentation of expressed antigen to CD8^+^ T cells [55]. Since GP5 could interact with antigen presentation related proteins (Figure 5), we assumed that GP5 might block the antigen presentation process to affect the immune response. Therefore, we examined the impact of PRRSV infection and GP5 over-expression on the expression levels of identified antigen presentation related proteins. Results showed that PRRSV infection down-regulated the expression levels of TAP1, TAP2, CANX, CALR, SLA-DMA, and SLA-DRA; furthermore, both PRRSV infection and GP5 over-expression significantly decreased the protein levels of SLA-II (Figure 6). The function of MHC-II is to bind and display peptides derived from exogenous or endogenous antigens for recognition by CD4^+^ T cells [49]. Previous studies have indicated that viruses such as vaccinia virus and HIV-1 could target MHC-II to restrict antigen processing and presentation pathways, ultimately lead to immune evasion [56,57,58]. Hence, we suspected that GP5 may be a key protein that inhibits the antigen processing and presentation pathway during PRRSV infection. It provides a new idea for exploring the mechanism of lower NAs levels induced by GP5, but further study should be conducted to confirm this hypothesis. 

Taken together, through comprehensive analysis and validation of GP5-interacting proteins, we elucidated the potential role of GP5 in antigen processing and presentation during PRRSV infection, which provides valuable reference for PRRSV pathogenesis and vaccine research and development.

## 4. Materials and Methods

### 4.1. Cells, Virus, and Reagents

HEK293T (ATCC CRL-11268) and MARC-145 cells (CCLV-RIE 0277) were preserved in our laboratory [59], which were cultured in Dulbecco Modified Eagle Medium (DMEM;) supplemented with 10% fetal bovine serum (FBS) at 37 °C with 5% CO_2_. Primary PAMs were collected via the bronchoalveolar lavage method from a four-week-old specific pathogen-free swine as previously described [59]. Briefly, piglets were euthanized, and lungs were obtained and washed with warm phosphate-buffered saline (PBS) containing 100 IU/mL penicillin and 100 μg/mL streptomycin at a sterile environment; then, the primary PAMs were collected from the lung lavage fluid. The cells were collected by centrifugation for 10 min at 400× *g* and cultured in RPMI Medium 1640 containing 10% FBS, 100 IU/mL penicillin, and 100 μg/mL streptomycin at 37 °C with 5% CO_2_. PRRSV Hn07-1 (a Type 2 PRRSV, GenBank accession no. KX766378) was generated in MARC-145 cells. 

Anti-Flag rabbit monoclonal antibodies (mAb; 14793) were purchased from Cell Signaling Technology (Boston, MA, USA). Anti-Myc rabbit polyclonal antibodies (pAb; R1208-1) and anti-beta actin rabbit mAb (ET1702-67) were purchased from HuaAn Biotechnology (Hangzhou, China). Fluorescein isothiocyanate (FITC)-labeled goat anti-mouse IgG antibody (172-1806), horseradish peroxidase (HRP)-labeled goat anti-mouse IgG antibody (074-1806), and HRP-labeled goat anti-rabbit IgG antibody (074-1506) were purchased from Kirkegaard & Perry Laboratories Inc. (Shanghai, China). Alexa Fluor 546 donkey anti-rabbit IgG antibody (A11040), Lipofectamine® 2000 (11668019), coverslips (12-545-80), and PageRuler Prestained Protein Ladder (26617) were purchased from Thermo Fisher Scientific Inc. (Waltham, MA, USA). Anti-Flag M2 mouse mAb (F1804) and anti-Flag® M2 affinity gel (A2220) were purchased from Sigma-Aldrich (St. Louis, MO, USA). Anti-PRRSV-N mouse mAb, anti-PRRSV-GP5 mouse mAb, and anti-SLA-II rabbit pAb were prepared by our laboratory (Zhengzhou, China). NP-40 Lysis Buffer (P0013F), Phenylmethanesulfonyl fluoride (PMSF; ST506), Protein A/G Agarose (P2005), and Fast Silver Stain Kit (P0017S) were purchased from Beyotime Biotechnology (Shanghai, China). DMEM (12110), RPMI Medium 1640 (31800), Tween 20 (T8220), Triton X-100 (T8200), 4’,6-diamidino-2-phenylindole (DAPI; C0065) and Antifade Mounting Medium (S2100) were purchased from Solarbio^®^ Life Sciences (Beijing, China). FBS (FSP500) was purchased from ExCell Bio (Shanghai, China). Penicillin-Streptomycin Solution (abs9244) was purchased from Absin Bioscience (Shanghai, China). 4% paraformaldehyde (G1101) was purchased from Servicebio (Wuhan, China). Nitrocellulose blotting membrane (10600001) was purchased from Cytiva (Washington, DC, USA). TRIzol reagent (R401-01), HiScript II 1st Strand cDNA Synthesis Kit (R211-01), Phanta Max Super-Fidelity DNA Polymerase (P505-d1), Rapid Taq Master Mix (P222-01), and FastPure Gel DNA Extraction Mini Kit (DC301-01) were purchased from Vazyme Biotech Co., Ltd (Nanjing, China). 

### 4.2. Plasmid Construction

GP5, cloned from the PRRSV Hn07-1 genome, were constructed into pCMV-Myc-N vector (Clontech, Palo Alto, CA, USA, 631604). TAP1, TAP2, CANX, CALR, SLA-II, PSMB7, VPS16, and VPS18, cloned from the PAMs genome, were constructed into pCMV-Flag-N vector (Clontech, 635688). All plasmid constructs were confirmed by sequencing. The primer sequences used for plasmid construction are listed in Appendix A. 

### 4.3. Cell Transfection and Viral Infection

For cell transfection experiments, HEK293T or MARC-145 cells were seeded in designated plates or glass coverslips at a suitable seeding density. Upon reaching 70–80% confluence, cells were transfected with the indicated recombinant vectors using Lipofectamine® 2000 according to the manufacturer’s instructions. Subsequently, they were subjected to western blotting or IFA analysis. Regarding cell infection experiments, PAMs were inoculated with PRRSV at an MOI of 1 for 2 h, normal cells were used as a mock trial, and then the medium was replaced with the fresh RPMI-1640 medium containing 10% FBS. The cells were harvested for subsequent analysis at the indicated times.

### 4.4. Co-IP Assay for LC-MS/MS Analysis

To analyze the potential host proteins interacting with GP5, PAMs (4.5 × 10^7^ cells) were infected with PRRSV at an MOI of 1 for 36 h and then lysed using NP-40 lysis buffer with containing the PMSF (1 mM). Cell lysates were incubated with anti-PRRSV-GP5 mAb at 4 °C overnight and then incubated with protein A/G agarose for 4 h at 4 °C. After centrifugation, the supernatant was removed, and the pellets were suspended in a washing buffer. The centrifugation and resuspension processes were performed five times. Finally, the pellets were lysed in a lysis buffer. After SDS-PAGE electrophoresis, the gel was revealed by silver staining, and gel fragments were excised and identified by LC-MS/MS. LC-MS/MS were performed at Applied Protein Technology Biotechnology Co., Ltd. (Shanghai, China). 

### 4.5. Sample Processing for TMT-Based Proteomics

To explore the proteomic characterization of PRRSV infection, PAMs were infected with PRRSV at an MOI of 1 for 24 h, the cells of Mock group and PRRSV group were dissolved in lysis solution with SDT buffer (4% (*w*/*v*) SDS, 100 mM Tris-HCl pH 7.6, 0.1 M DTT) to extract the protein (n = 3). The TMT-based quantitative proteomic were performed at Applied Protein Technology Biotechnology Co., Ltd. (Shanghai, China) [4]. 

### 4.6. LC-MS/MS Data Analysis 

In this study, the original MS data were searched through the Mascot (version 2.3), and then the retrieved results were imported into the Proteome Discoverer (version 3.1) for the identification and quantitation analysis, and the quantification of a protein was calculated by the median of only the unique peptides of the protein. The Sus scrofa database was downloaded from UniProtKB (https://www.uniprot.org, accessed on 16 April 2023), and a false discovery rate (FDR) < 1% was used for the identification. We also searched against the contaminant database, the Common Repository of Adventitious Proteins (cRAP), during data analysis. For Co-IP-LC-MS/MS, to reduce the probability of false peptide identification, each confident protein involved at least two unique peptides, and peptides with average Mascot scores ≥14 were counted as identified. For TMT-Based Proteomics, the fold changes were equal to the ratio of the PRRSV infection group values over the Mock group values.

### 4.7. Bioinformatics Analysis

GO analysis and the KEGG pathway annotation of all the GP5-interacting proteins were performed using DAVID Bioinformatics Resources (http://david.ncifcrf.gov, accessed on 8 November 2023). The enrichment analysis of these GP5-interacting proteins for the GO annotation and KEGG pathway annotation were performed based on Fisher’s exact test (*p* < 0.05). Moreover, the interaction relationship between target proteins was based on the STRING database (https://cn.string-db.org, accessed on 11 November 2023), and the result was visualized by Cytoscape (version 3.7.0). The PPI network was generated with confidence score > 0.7 (high confidence). 

### 4.8. Co-IP Assay 

HEK293T cells cultured in 90 mm dishes (6 × 10^6^ cells) were separately co-transfected with the indicated recombinant vectors by Lipofectamine^®^ 2000 Transfection Reagent. After 24 h, cells were lysed using NP-40 lysis buffer with containing the PMSF (1 mM). Cell lysates were incubated with anti-Flag M2 affinity gel for 4 h at 4 °C. After centrifugation, the supernatant was removed, and the pellets were suspended in a washing buffer. The centrifugation and resuspension processes were performed five times. Finally, the pellets were lysed in a lysis buffer for western blotting analysis.

### 4.9. Western Blotting

Cells were harvested and lysed immediately in lysis buffer (2% SDS, 1% Triton X-100, 50 mM Tris-HCl, 150 mM NaCl, pH 7.5). Different proteins were separated using SDS-polyacrylamide gel electrophoresis, and the separated protein bands were blotted onto a nitrocellulose blotting membrane. After blocking the membrane with 5% nonfat dry milk containing 0.1% Tween 20 for 30 min at 37 °C, the membranes were incubated with primary antibodies for 6 h at 4 °C, followed by incubation with HRP-labeled anti-mouse/rabbit IgG for 1 h at 37 °C, after which the membranes were visualized using Amersham Image Quant 800 (Cytiva, Washington, DC, USA).

### 4.10. Immunofluorescence

MARC-145 cells cultured on coverslips in 12-well plates (1 × 10^5^ cells) were separately co-transfected with the indicated vectors by Lipofectamine^®^ 2000 Transfection Reagent. After 36 h, the cells were fixed with 4% paraformaldehyde for 10 min at room temperature and then blocked with 5% nonfat milk for 1 h. After washing the cells thrice with phosphate-buffered saline with Tween 20 (PBST), they were incubated with the indicated primary antibodies. The cells were then washed with PBST and incubated with fluorescent-conjugated IgG antibodies. After washing the cells thrice with PBST, they were incubated with DAPI. The cells were washed with PBS and then observed under the Zeiss laser scanning confocal microscope (Zeiss, Oberkochen, Germany). 

### 4.11. Statistical Analysis

The protein bands were relatively quantified from western blotting analysis. Briefly, the mean gray value of protein bands within the linear range and the background were measured using ImageJ software (version 1.46), and the quantification values will reflect the relative amounts as a ratio of each net band value relative to the net loading control. Data are presented as mean ± SD from three independent samples. Statistical analysis was performed by GraphPad Prism software (version 8.0) with Student’s *t*-test. *p* < 0.05 indicated statistical significance.

## Figures and Tables

**Figure 1 ijms-25-02778-f001:**
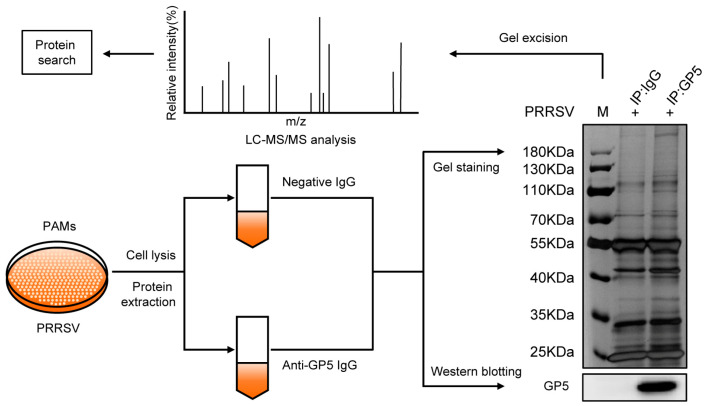
IP-MS analysis of GP5-interacting proteins. PAMs were infected with PRRSV for 36 h and then harvested for Co-IP. The resultant cell lysates were subjected to immunoblotting using the indicated antibody. Meanwhile, after SDS-PAGE electrophoresis, the gel was revealed by silver staining, and the gel fragment was excised to identified by LC-MS/MS.

**Figure 2 ijms-25-02778-f002:**
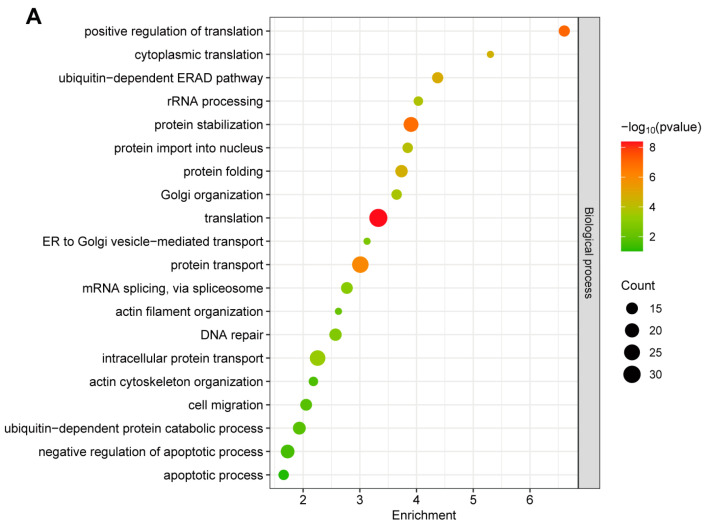
GO annotation of GP5-interacting proteins. (**A**) GO enrichment analysis for biological process. (**B**) GO enrichment analysis for cellular component. (**C**) GO enrichment analysis for molecular function. The Top 20 enriched terms were revealed. The abscissa in the figure is the enrichment. The color of the dot represents the *p* value of the hypergeometric test. The color ranges from green to red. The redder the color is, the smaller the value is. The size of the point represents the number of proteins.

**Figure 3 ijms-25-02778-f003:**
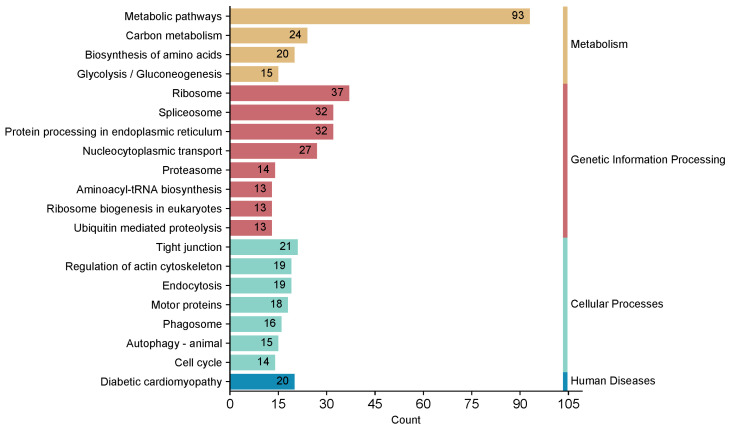
KEGG pathway annotation of GP5-interacting proteins. The Top 20 enriched pathway terms were revealed.

**Figure 4 ijms-25-02778-f004:**
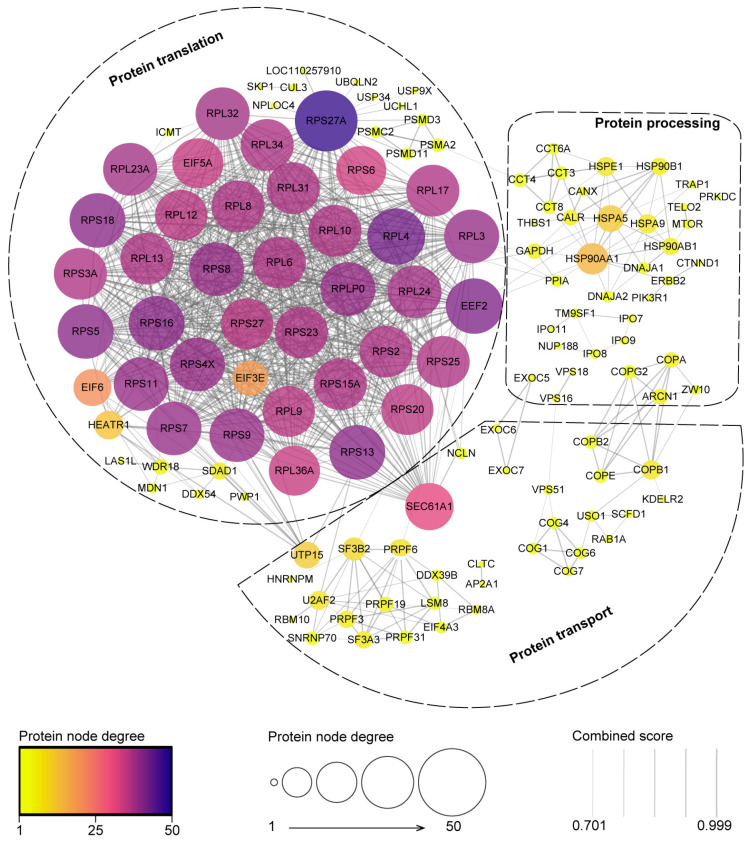
PPI analysis of GP5-interacting proteins. The GP5 interaction network was built from the STRING database with confidence score > 0.7 (high confidence). These proteins were mainly concentrated in three PPI network clusters, including protein translation, protein transport, and protein processing, which were, respectively, circled with three different shapes.

**Figure 5 ijms-25-02778-f005:**
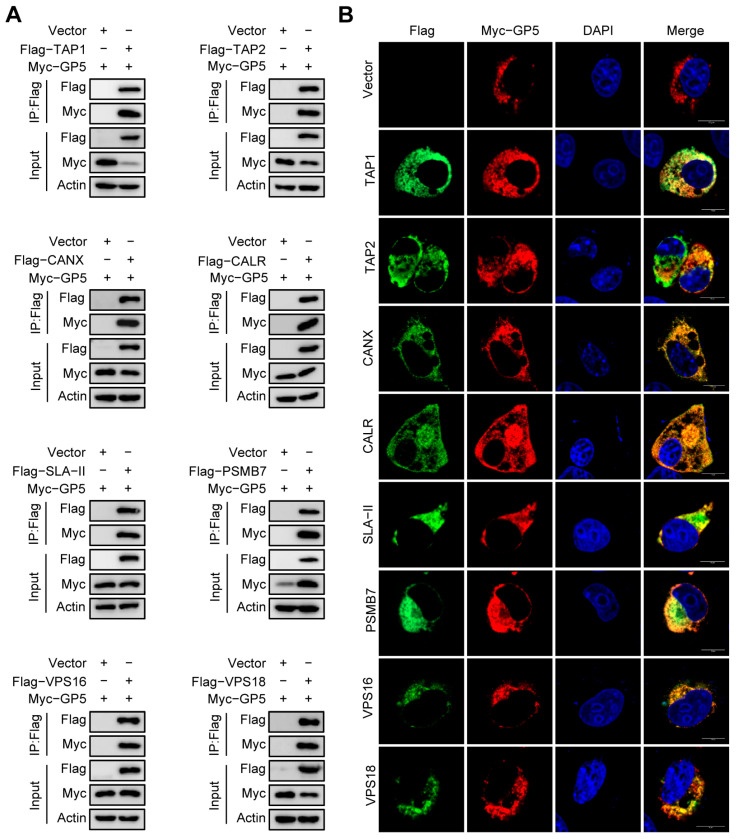
Identification of the selected GP5-interacting proteins. (**A**) HEK293T cells were co-transfected with Flag-vector encoding the indicated host proteins and Myc-GP5 for 24 h. The cell lysates were immunoprecipitated with anti-Flag IgG beads and subjected to immunoblotting using the indicated antibody. (**B**) MARC-145 cells were co-transfected with Flag-vector encoding the indicated host proteins and Myc-GP5 for 36 h. Cells were immunostained and observed using confocal microscopy. Scale bar indicates 10 μm.

**Figure 6 ijms-25-02778-f006:**
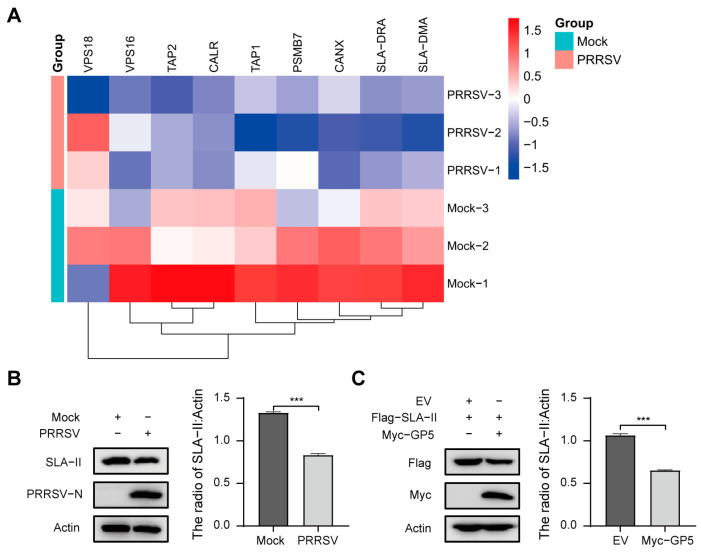
The effects of PRRSV and GP5 on expression levels of identified proteins. (**A**) PAMs were infected with PRRSV at an MOI of 1 for 24 h, and cells were harvested for a TMT-based quantitative proteomic analysis. (**B**) PAMs were infected with PRRSV at an MOI of 1 for 24 h, and the cell lysates were subjected to immunoblotting using the indicated antibody. (**C**) HEK293T cells were co-transfected with Flag-SLA-II and Myc-GP5 for 24 h. The resultant cell lysates were subjected to immunoblotting using the indicated antibody. EV: Empty vector. The protein bands were relatively quantified by ImageJ software (version 1.46). Error bars: mean ± SD of 3 independent tests. Student’s *t*-test; *** *p* < 0.001 compared to control.

## Data Availability

All the data were included in the manuscript and Appendix A. The original data reported in this paper have been deposited in the OMIX, China National Center for Bioinformation/Beijing Institute of Genomics, Chinese Academy of Sciences [https://ngdc.cncb.ac.cn/omix: accession no. OMIX005809 (Co-IP-LC-MS/MS) and OMIX002135 (TMT-Based Proteomics)].

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
