# Peer review of "Mass Spectrometry-Based Proteomic Analysis of Potential Host Proteins Interacting with GP5 in PRRSV-Infected PAMs"

_ijms, 2024, doi:10.3390/ijms25052778_

Round 1

Reviewer 1 Report

Comments and Suggestions for Authors

The study of Li and coauthors is aimed at identifying porcine proteins interacting with the GP5 protein of a major viral pathogen. There is an indisputable novelty, the paper is generally well organized and written.

My only major concern relates to selection of LC-MS/MS-identified proteins for bioinformatic analysis. A part of 720 amino acid sequences listed in Table S1 could belong to regulatory peptides and distinct protein forms, and therefore be not employed in inferred pathways. Please consider the issue in the main text. Many proteins were identified by two or three peptides. How reliable are these results? Is it possible to provide identification score values? There should be an explicit justification of inclusion of proteins identified by few peptides into the pathway analysis, supported by relevant citations. It may require re-analysis of the data on the basis of some established threshold, not just a threshold of two and more found peptides.

Minor comments

I wonder why neither primary nor secondary macrophage receptors binding PRRV (CD163 and CD169) were found [1]. Please look for these proteins in the data and discuss their presence or absence in the paper.

The name of the virus family Arteriviridae should be spelled in italics

Figure 1. Fix a typo in “negative”

[1] Crisci E, Fraile L, Montoya M. Cellular Innate Immunity against PRRSV and Swine Influenza Viruses. Vet Sci. 2019 Mar 11;6(1):26. doi: 10.3390/vetsci6010026. PMID: 30862035.

Reviewer 2 Report

Comments and Suggestions for Authors

This study uses co-immunoprecipitation, mass-spectrometry and bioinformatics to identify porcine alveolar macrophage host proteins that interact with porcine reproductive and respiratory syndrome virus GP5 during infection. Several proteins within the antigen processing and presentation pathway are then further investigated.

General Comments

This manuscript would benefit from an English language editor.

The introduction does not lay out a specific question for the study. Therefore, the support/background for the objective is vague and weak.

Methods should be written to include all pertinent information for each experiment. It is very difficult to understand as it is currently written.

Abstract

Line 10 - PRRSV has a large economic impact on the swine industry. Saying PRRSV is "destroying the global swine industry" is an overstatement of the situation.

Line 16+ "In-depth" is not needed. The main result of the study appears to be the interaction of GP5 with antigen processing and presentation pathways in PAMs. But this finding is only mentioned on Line 21.

Line 21 - Based on the analyses and results presented in the study, saying that "PRRSV may utilize GP5 to destroy the antigen processing and presentation pathway" is an overstatement. The experiments were not done in this study.

Introduction

Line 45 - More detail about what is currently known about GP5's interactions with host proteins is needed.

Line 47 - Studying GP5 is important for diagnosis and vaccine development. But how will understanding interactions of GP5 with host proteins lead to prevention and control of PRRSV?

Line 51 - Specifically, how will understanding will GP5-host interactions advance control of viral infection?

Line 52 - It is important to describe why GP5-host protein interactions in MARC-145 cells may be different from PAMs. It would also be helpful to the reader to briefly explain what role PAMs play in the immune response to PRRSV infection.

Results

Line 65 - Shouldn't this be co-immunoprecipitation?

Figure 3 - What is intensiveness?

Line 113 - How were the 163 proteins chosen?

Line 127 - The details associated with the antigen processing and presentation pathway in the KEGG data needs to be described. All of the GO and KEGG results presented does not include antigen processing and presentation. It makes it hard for the reader to understand why these proteins were chosen for further analysis.

Line 129 - How were the 8 proteins chosen? What are their proposed functions?

Line 134 - What is the distribution of GP5 and the selected proteins in infected PAMs? How do they compare to what is shown by over-expression? Is the GP5 sequence in the expression vector only the ectodomain? How would expression of a full-length GP5 change its localization?

Line 145 - How does the tandem mass-tag work differently from LC-MS/MS to give quantitative results? How are fold changes calculated for this assay?

Figure 6B - Why is SLA-DRA the only target shown?

Figure 6C - Is Vector representing the empty vector or the SLA-II expression vector?

Discussion

Line 170 - NA has not been defined.

Line 180 - This sentence sounds like bioinformatic analyses were done on GP5 rather than the proteins identified by LC-MS/MS.

Line 196-197 - This sentence seems out of place. These proteins are not mentioned in the text and there is no comparison of the proteins with those found in the current study.

Line 198 - As mentioned earlier, there is no data presented that shows the antigen processing and presentation pathway was found in the GO or KEGG analyses. Wouldn't this result also be more likely found in cells, such as macrophages, that antigen presenting cells?

Line 202 - What does "main protective antigenic protein" mean?

Line 205 - Some speculation on how the inhibition of these proteins by PRRSV would impact the immune response to infection would be appropriate.

Line 236 - Destroy is an overstatement based on the data presented.

Comments on the Quality of English Language

The manuscript would benefit from an English language editor.

Reviewer 3 Report

Comments and Suggestions for Authors

The authors did not deposit MS data to proteomic database such as ProteomeXchange Consortium https://www.proteomexchange.org/. Please submit MS data generated during the study and provide dataset identifier number in the revised manuscript.

Methods and materials: It is not mentioned anywhere in the manuscript that how many samples were used and analysed.

Lines 305-3011, did you use cRAP (common Repository of Adventitious Proteins) during data analysis? Additionally, authors did not provide any information on database searches such as name of whole genome and accession number along with taxonomy id and entries.

Lines 348-349: what was the RIN values of RNA samples? What was the RNA concentration to synthesize first cDNA.

Table S4: Add PCR product size for each primer set

Round 2

Reviewer 1 Report

Comments and Suggestions for Authors

The requested corrections were carefully implemented. I have no further comments.

Comments on the Quality of English Language

/

Reviewer 3 Report

Comments and Suggestions for Authors

The authors provided weblink https://ngdc.cncb.ac.cn/ and accession numbers. But I could not find these accession numbers in this database. I would like to see these generated data in the data bank. Please send provide a screenshot or reviewer access link.

I do not believe that RIN value of RNA samples is correctly examined by agarose gel electrophoresis. The authors must use sophisticated instruments.

Round 3

Reviewer 3 Report

Comments and Suggestions for Authors

The authors have made all the necessary edits. The manuscript now appears much better readable.

Author Response

Dear Reviewer:

Thank you for your positive and constructive comments regarding our manuscript, entitled “Mass spectrometry-based proteomic analysis of potential host proteins interacting with GP5 in PRRSV-infected PAMs” (Manuscript ID: ijms-2831735). They are all valuable and very helpful for revising and improving our paper. Once again, we are very grateful for your professional reviews, constructive comments, and valuable suggestions on our manuscript.